# Tamarind Xyloglucan Oligosaccharides Attenuate Metabolic Disorders via the Gut–Liver Axis in Mice with High-Fat-Diet-Induced Obesity

**DOI:** 10.3390/foods12071382

**Published:** 2023-03-24

**Authors:** Chun-Hua Zhu, Yan-Xiao Li, Yun-Cong Xu, Nan-Nan Wang, Qiao-Juan Yan, Zheng-Qiang Jiang

**Affiliations:** 1Department of Nutrition and Health, College of Food Science and Nutritional Engineering, China Agricultural University, Beijing 100083, China; 2Key Laboratory of Food Bioengineering (China National Light Industry), College of Engineering, China Agricultural University, Beijing 100083, China; 3College of Food Science and Engineering, Collaborative Innovation Center for Modern Grain Circulation and Safety, Nanjing University of Finance and Economics, Nanjing 210023, China

**Keywords:** tamarind xyloglucan oligosaccharides, gut microbes, obesity, gut–liver axis

## Abstract

Functional oligosaccharides exert obesity-reducing effects by acting at various pathological sites responsible for the development of obesity. In this study, tamarind xyloglucan oligosaccharides (TXOS) were used to attenuate metabolic disorders via the gut–liver axis in mice with high-fat-diet (HFD)-induced obesity, as determined through LC/MS-MS and 16S rRNA sequencing technology. A TXOS dose equivalent to 0.39 g/kg/day in humans restored the gut microbiota in obese mice, which was in part supported by the key microflora, particularly *Bifidobacterium pseudolongum*. Moreover, TXOS reduced the abundance of opportunistic pathogen species, such as *Klebsiella variicola* and *Romboutsia ilealis.* The bodyweight and weight gain of TXOS-treated (4.8 g/kg per day) mice began to decrease at the 14th week, decreasing by 12.8% and 23.3%, respectively. Sixteen fatty acids were identified as potential biomarkers in the liver, and *B. pseudolongum* and caprylic acid were found to tightly regulate each other. This was associated with reduced inflammation in the liver, circulation, and adipose tissue and protection from metabolic disorders. The findings of this study indicate that TXOS can significantly increase the gut microbiota diversity of obese mice and restore the HFD-induced dysbiosis of gut microbiota.

## 1. Introduction

Obesity causes many metabolic disorders and is associated with the accumulation of fat tissue, weight gain, gut microbiota dysbiosis, chronic inflammation, metabolic endotoxaemia, and nonalcoholic fatty liver disease [1,2]. The global prevalence of obesity will reach more than 21% in women and 18% in men by 2025 [3].

Prebiotics mainly include functional oligosaccharides and dietary fibres, which show obesity-reducing effects by acting at various pathological sites responsible for the development of obesity [4,5]. Fructooligosaccharides, galactooligosaccharides, lactulose, and nondigestible fibres such as inulin, cellulose, resistant starch, hemicellulose, gum, and pectin are the most commonly used prebiotics in practice [6]. Several mechanisms that specifically promote the growth of inherited gut microbes or probiotics have been reported in animal experimental models [7]. Moreover, prebiotics enhance short-chain fatty acid (SCFA) production by gut bacteria, alter the composition of SCFAs, and affect the bile acid (BA) pathway [8,9]. The gut microbiota is a promising therapeutic target for obesity [10,11]. Xyloglucan, as one kind of hemicellulose, can reduce liver damage [12] and exhibits anti-inflammatory [13] and cholesterol-lowering activities [14]. Microbes from alginate oligosaccharide-treated mice successfully mitigated small intestinal mucositis [15]. Treatment with the odd-numbered oligosaccharides from Gracilaria agarose (OGAOs) alleviated obesity induced by a high-fat diet (HFD) and increased the relative abundance of *Akkermansia* [16]. Mannoligosaccharides restored the composition of gut microbiota, increasing the relative abundance of *Lactobacillus* and reducing the relative abundance of *Helicobacter* [17]*. Proteobacteria*, *Verrucomicrobia*, and *Actinobacteria* are associated with histological severity among obese patients [18]. Based on crosstalk between the liver and microbiota, a ‘liver–microbiome’ axis has been proposed, with implications for health and disease. The gut microbiota and its metabolites modulate liver gene expression directly and indirectly, causing an imbalance in the gut–liver axis, which may result in disease, including carcinogenesis [19]. The intestinal microbiota control liver and metabolic functions [20,21]. In turn, the liver impacts the microbiota through hepatic mediators, such as bile acids or inflammatory signals, and communicates with it. Therefore, the liver–microbiome bidirectional crosstalk appears to be critical for health and in various liver diseases, and may be therapeutically targeted [22].

Tamarind (*Tamarindus indica* L.) seeds are rich in xyloglucan. This substance aggregates in cross-like and parallel-like assemblies, which may have rope-like structures [23]. In our previous study, tamarind xyloglucan oligosaccharides (TXOS) with a degree of polymerization (DP) of 7–9 were prepared by using xyloglucanase (RmXEG12A) from tamarind powder [24]. Tamarind seed polysaccharides have potential prebiotic properties, including anti-obesity and anti-inflammation properties [25]. However, whether TXOS can restore the dysbiosis of the gut microbiota induced by HFD remains unknown. In this study, we investigated the intervention and regulatory effects of TXOS on the gut microbiota of obese mice.

## 2. Materials and Methods

### 2.1. Animals and Treatments

Animal experiments were performed in accordance with the Guidelines for Animal Experimentation of China Agricultural University (Beijing, China), and the protocols were approved by the Animal Ethics Committee (approval number:20185001-3). The animal breeding licence number was SCXK(Jing)2016-0011. A total of 60 male C57BL/6J mice (17–20 g) were purchased from Beijing Vital River Laboratory Animal Technology Co., Ltd. (Beijing, China). The mice were bred in an animal room under a cycle of 12:12 h day/night with water and chow provided ad libitum. Tamarind powder was hydrolysed by the xyloglucanase RmXEG12A to prepare xyloglucan oligosaccharides [24]. Briefly, 30.0 g tamarind powder was dispersed in 200 mL MOPS buffer pH 7.0 (50 mM) under magnetic stirring. RmXEG12A (1.5 mL) with a protein concentration of 1.3 mg mL^−1^ was added, and an enzymatic reaction was carried out at 55 °C for 12 h with a constant rotation speed of 160 rpm. The hydrolysates were immediately boiled for 20 min and centrifuged at 10,000× *g* for 10 min to collect the supernatants. Thereafter, the supernatants were lyophilized and stored at 25 °C. For the animals’ treatment, the 60 mice were randomly divided into six groups, 10 animals per group, after feeding for one week. Mice in the control group were fed a normal diet (ND) with 10 kcal% fat (Beijing Keao Xieli Feed Co., Ltd., Beijing, China), while mice in the negative control (HFD, orally treated with water), HFDA group (orally treated with 15.6 mg/kg per day orlistat), and three treatment groups (orally treated with three doses of TXOS viz. 1.6, 3.2, and 4.8 g/kg per day, which corresponded to the three groups HFDL, HFDM, and HFDH) were supplied an HFD with 60 kcal% fat (D12492, Beijing Yicheng Technology Co., Ltd., Beijing, China) for 19 weeks (Appendix A). Mice in the HFDA group were used as an anti-obesity drug positive control. Bodyweight and feed intake were assessed every week during the experimental period. Energy intake was analysed according to the dietary intake (g) and the energy level of the diet (Appendix A).

### 2.2. RNA Extraction and Quantitative PCR Analysis (qRT-PCR)

Total RNA was extracted from the mouse livers and epididymal fat using TRIzol reagent (Thermo Fisher Scientific, Carlsbad, CA, USA)) and then reverse-transcribed into cDNA using the PrimeScript RT (Master Mix RR036A, TAKARA, Dalian, China) Reagent kit. The gene expression levels were quantified using reverse transcription quantitative PCR (RT-qPCR) (CFX96, Bio-Rad Laboratories, Hercules, CA, USA) with the primers listed in Appendix A. The PCR conditions were as follows: 95 °C for 30 s; 40 cycles of amplification (95 °C for 5 s, 60 °C for 30 s); and 72 °C for 30 s. Then, melting curves were acquired stepwise from 55 to 95 °C. Expression levels were normalized to that of glyceraldehyde-3-phosphate dehydrogenase (GAPDH). The expression levels of target genes were measured using the comparative cycle threshold method.

### 2.3. Biochemical Tests

The enzyme activities of superoxide dismutase (SOD), malonyldialdehyde (MDA), glutathione peroxidase (GSH-Px), and catalase (CAT) in the liver were measured using kits from Nanjing Jiancheng Bioengineering Institute (Nanjing, Jiangsu, China) according to the manufacturer’s protocols. The high-density lipoprotein cholesterol (HDL-C), low-density lipoprotein cholesterol (LDL-C), total cholesterol (TC), and triglyceride levels in plasma were determined using an automatic biochemical analyser (IChem-340) from Shenzhen Kubel Biotechnology Co., Ltd. (Shenzhen, Guangdong, China).

### 2.4. Histological Analysis

Samples of liver tissues were placed in 4% PBS–paraformaldehyde solution, embedded in paraffin solution, sectioned, and stained with haematoxylin and eosin (H&E). Frozen mouse liver pieces were used for oil red O staining and quantified using a Nikon Ci-S microscope equipped with a digital camera (DS-U3, Nikon Corp., Konan, Minato-ku, Tokyo).

### 2.5. Fatty Acid Analysis

All fatty acid standards (47 + 5) were purchased from Shanghai Yuanye Biological Co., Ltd., and they were chromatographically pure. A total of 47 fatty acid standards and 5 stable isotope-labelled standards were obtained from ZZ Standards Co., Ltd. (Shanghai, China). Isopropanol (Optima LC–MS), acetonitrile (Optima LC–MS), and formic acid (Optima LC–MS) were purchased from Thermo-Fisher Scientific (Fairlawn, NJ, USA). Ultrapure water was prepared by Millipore (Burlington, MA, USA). The stock solution of individual fatty acids was mixed and prepared in a fatty acid-free matrix to obtain a series of fatty acid calibration solutions with concentrations of 40,000, 20,000, 10,000, 4000, 2000, 1000, 400, 200, 100, 40, 20, and 10 ng/mL. Solutions with certain concentrations of decanoic acid-d19, myristic acid-d2, octadecanoic acid-d35, eicosanoic acid-d39, and lignoceric acid-d4 were prepared and mixed in as internal standards. The stock solutions of all of these solutions and the working solutions were stored in a freezer at −20 °C. The samples (100 mg) were resuspended in liquid nitrogen, homogenized with 300 μL of isopropanol/acetonitrile (1:1), which contained the mixed internal standards, and centrifuged at 12,000 rpm for 10 min. The supernatant (2 μL) was injected into an LC–MS/MS system for analysis. An ultrahigh-performance liquid chromatography coupled with tandem mass spectrometry (UHPLC–MS/MS) system (ExionLC™ AD UHPLC-QTRAP 6500+, AB SCIEX Corp., Boston, MA, USA) was used to quantitate the fatty acids. Separation was performed on a Waters ACQUITY UPLC BEH C18 column (2.1 × 100 mm, 1.7 μm) maintained at 50 °C. The mobile phase, consisting of 0.05% formic acid in water (solvent A) and isopropanol/acetonitrile (1:1) (solvent B), was delivered at a flow rate of 0.30 mL/min. The solvent gradient was set as follows: 30% B for 1 min, followed by 30–65% B for 2 min, 65–100% B for 11 min, 100% B for 13.5 min, 100–30% B for 14 min, and 30% B for 15 min. The mass spectrometer was operated in negative multiple reaction monitoring (MRM) mode. The parameters used were as follows: ion spray voltage, −4500 V; curtain gas, 35 psi; ion source temperature, 550 °C; and ion source gas 1 and 2, 60 psi.

### 2.6. DNA Extraction and 16S rRNA Gene Sequencing of Mouse Faeces

The mouse faecal samples were placed in cryopreservation tubes, which were quickly placed in liquid nitrogen and stored in a freezer at −80 °C for later use. DNA was extracted from the faeces with a DNA extraction kit. The quality of DNA samples was checked through agarose gel electrophoresis. After agarose gel electrophoresis, the DNA samples were stored at −20 °C. The sequencing region was 338f–806r (338f: actcctacgggaggcagcagcag; 806r: ggactachvgggtwtctaat). PCR was carried out using TransStart FastPfu DNA polymerase (Transgenic, ap221-02) in a 20-μL reaction system, which contained 4 μL of 5 × FastPfu Buffer, 2 μL of 2.5 mM dNTPs, 0.8 μL of forward primer (5 μM), 0.8 μL of reverse primer (5 μM), 0.4 μL of FastPfu polymerase, 0.2 μL of BSA, 10 ng of template DNA, and ddH2O up to 20 μL. PCR was performed as follows: 1× (3 min at 95 °C), cycle number × 3 (30 s at 95 °C; 30 s at the annealing temperature of 85 °C; and 45 s at 72 °C), 1× (10 min at 65 °C), and 10 °C until halted by the user. The DNA samples were fragmented using a Covaris M220 device (Gene Company, Shanghai, China) and screened for fragments of approximately 400 bp. A NEXTFLEX Rapid DNA-Seq Kit (Bio Scientific, USA) was used to construct the paired-end sequencing library. The V3–V4 regions of the 16S ribosomal RNA (16S rRNA) gene were amplified, and sequencing was conducted on an Illumina NovaSeq platform (Illumina, San Diego, CA, USA) at Novogene Technology Co., Ltd. (Beijing, China).

Sequencing libraries were generated using a NEB Next^®^ Ultra™ DNA Library Prep Kit for Illumina (NEB, Ipswich, MA, USA) following the manufacturer’s recommendations, and index codes were added. The library quality was assessed on the Qubit@ 2.0 Fluorometer (Thermo Scientific) and Agilent Bioanalyzer 2100 system. Finally, the library was sequenced on an Illumina HiSeq platform and 250 bp paired-end reads were generated. Paired-end reads from the original DNA fragments were merged by using FLASH—a very fast and accurate analysis tool that is designed to merge paired-end reads when there are overlaps between reads1 and reads2. Paired-end reads were assigned to each sample according to the unique barcodes. Sequences were analysed using the QIIME software package (Quantitative Insights into Microbial Ecology), and in-house Perl scripts were used to analyse alpha (within samples) and beta (among samples) diversity. First, reads were filtered using QIIME quality filters. Then, we used pick_de_novo_otus.py to pick operational taxonomic units (OTUs) by making an OTU table. Sequences with ≥97% similarity were assigned to the same OTUs. We picked representative sequences for each OTU and used the RDP classifier to annotate taxonomic information for each representative sequence.

### 2.7. Statistical Analysis

The results were analysed using IBM SPSS 23.0 and are presented as the means ± standard deviations (SDs) of three replicates. A *t*-test was used to analyse the differences between two groups, and one-way ANOVA was used to compare the differences between more than two groups, followed by Tukey’s HSD post hoc test at * *p*-value < 0.05 and ** *p*-value < 0.01. Metabolomic data were analysed using partial least squares discriminant analysis (PLS-DA). Heatmap analysis and processing were performed with the web-based tools MetaboAnalyst 5.0 (http://www.metaboanalyst.ca/, accessed on 22 April 2022) and MetOrigin (http://metorigin.met-bioinformatics.cn, accessed on 22 April 2022). Figures were prepared with GraphPad Prism version 8.0 (San Diego, CA, USA) and the open-source web service Hiplot (https://hiplot.com.cn, accessed on 22 April 2022). Spearman’s correlation analysis was used for the correlation analysis.

## 3. Results

### 3.1. Effects of TXOS on Bodyweight and the Serum of Obese Mice

During the 19-week experimental period, the C57/BL 6J mice displayed a gradual increase in bodyweight (Table 1). The bodyweight and weight gain of the TXOS-treated (4.8 g/kg per day) mice in the HFDH group began to decrease at the 14th week, decreasing by 12.8% and 23.3%, respectively. Compared with the mice in the HFD group, the amounts of perirenal fat, epididymal fat, and subcutaneous fat in the HFDH group mice were reduced by 20.2%, 20.9%, and 3.8%, respectively. No significant difference in food intake was observed between the different groups. Thus, the high dose of TXOS (4.8 g/kg per day) effectively regulated adipose tissue accumulation in obese mice.

Serum lipid profile analysis showed that TXOS treatment decreased the level of serum triglycerides (Appendix A). Across the six groups, the high dose of TXOS significantly reduced the levels of TC and LDL-C in the mice (^##^ *p*-value < 0.01), and the inhibition rates were 23.1% and 35.9%, respectively. Moreover, the level of HDL-C was increased by approximately 1.0-fold.

### 3.2. Effect of TXOS on Adipose and Liver Tissues in Mice

As shown in Appendix A, the HFD group mice showed a significantly increased cell area of epididymal adipose tissue (* *p*-value < 0.05). TXOS treatment significantly reduced the area of adipose tissue cells and significantly increased the number of adipocytes (** *p*-value < 0.01) in the same visual field. Hepatic lobules were clearly observed in the ND group nice, and the hepatic cords were neatly arranged (Figure 1A,B). Compared with those in the ND group mice, the number and volume of hepatic fat vacuoles were significantly increased in the HFD group mice. The number and volume of hepatic fat vacuoles in obese mice were significantly reduced by three doses of TXOS. The results indicated that TXOS treatment significantly modulated the liver pathology of mice.

Further exploration of the expression of lipid metabolism-related genes showed that the key genes for adipogenesis were clearly expressed in the HFD group (Appendix A). These genes included genes encoding peroxisome proliferator-activated receptor (PPARG), cholesterol regulatory element-binding protein 1c (SREBP1c), and acetyl-CoA carboxylase (ACACA). High-dose TXOS treatment significantly downregulated the expression of PPARG, ACACA, and FASN (** *p* < 0.01) by 57.8%, 11.0%, and 15.2%, respectively. The MDA activity (3.19 nmol/mg) in the livers of the HFD group mice was much higher than that in the ND group mice (1.28 nmol/mg). The MDA activity in mouse livers was significantly decreased after TXOS treatment for 16 weeks (* *p*-value < 0.05) (Figure 1D). Moreover, the SOD activity was elevated by 8.5% in the HFDH group mice compared to that in the HFD group mice (* *p*-value < 0.05). These results indicated that the total antioxidant capacity of mouse livers was improved after treatment with TXOS. The mRNA expression levels of LXR, PPAR α, and GPNMB in the liver were determined (Appendix A). The mRNA expression of LXR was decreased in the three TXOS treatment groups compared with the HFD group. The mRNA level of PPAR α was increased only in the HFDH group at 19 weeks.

### 3.3. TXOS Supplementation Changed the Composition of Hepatic Fatty Acids

Forty-seven fatty acids were detected in the mouse livers after supplementation with TXOS (Appendix A). The OPLS-DA plot revealed a significant difference in the abundance of liver fatty acids between the different groups (Figure 2A). The results revealed a distinct clustering of microbial structure for each group, and the gut microbiota compositions of the HFDH group mice were similar to those of the HFDA group mice. As shown in Appendix A, a fold change ≥2.0 was used as the screening criterion.

A total of 16 potential biomarkers were identified (Table 2). Compared with those in the ND group, the levels of C16:1T, C18:1(n-12), C20:5, and C14:1T were significantly increased, while the levels of three biomarkers (C24:1, C8:0 and C21:0) were markedly decreased in the HFD group. The levels of 12 differentially abundant metabolites ((C24:0, C14:1T, C18:2(n-6), C18:2(n-6)T, C20:5, C18:3(n-3), C12:0, C14:0, C8:0, C16:1T, C13:0, and C22:5(n-3)) were notably regulated after TXOS treatment (Appendix A). The KEGG (https://www.genome.jp/kegg/, accessed on 22 April 2022) and MetPA (https://www.metaboanalyst.ca, accessed on 22 April 2022) databases were used for metabolic pathway analysis. Based on the 16 identified biomarkers, five metabolic pathways were found. Moreover, three metabolic pathways (lipoic acid metabolism, α-linolenic acid metabolism, and linolenic acid metabolism) were selected as the most important metabolic pathways (* *p*-value < 0.05, impact > 0.1), and were all related to metabolic disturbances (Figure 2B). The linkage of all the possible bacteria with linoleic and lipoic acid metabolism is shown in Figure 2C,D. For example, *Klebsiella variicola* might also affect linoleic acid via the lipoic acid pathway directly. *Bifidobacterium pseudolongum* and caprylic acid tightly regulate each other. Overall, we speculated that there was an important interplay between bacteria and metabolites after TXOS treatment in obese mice.

### 3.4. TXOS Alleviated Obesity Caused by HFD-Induced Gut Dysbiosis in Mice

To examine the effects of TXOS on gut microbiota composition, 16S rRNA sequencing of mouse faeces was performed. Partial least squares discrimination analysis (PLS-DA) revealed a significant difference in the abundance of OTUs between different groups (Figure 3A). The overlapping part in the middle (the number “340”) was the common OTUs found in all treatment groups (Figure 3B). Numbers without the overlap represent the number of unique OTUs of the group. The Shannon and Chao1 indexes are shown in Figure 3C,D, and there were enough sequencing data to reflect the species information of most microorganisms in the samples. Notably, the Firmicute/Bacteroidetes abundance ratio was decreased after TXOS treatment (Figure 3E). Variance analysis was further used to identify the specific bacterial phylotypes altered after TXOS treatment. TXOS significantly increased the abundance of 39 operational taxonomic units (OTUs) and decreased the abundance of 37 OTUs in a dose-dependent manner (Figure 4A). Most of these altered OTUs belonged to *Bacteroidales*, *Muribaculaceae* (25 OTUs); *Bifidobacteriales*, *Bifidobacteriaceae* (1 OTU); *Firmicutes*, *Lachnospiraceae* (11 OTUs); *Firmicutes*, *Clostridiaceae* (2 OTUs); and *Bacteroidales*, *Prevotellaceae* (2 OTUs).

The cladogram shows the dominant bacteria in each group (Figure 5A). The taxonomy at the phylum level was dominated by Actinobacteria and Firmicutes in all the sections. Other prominent phyla included *Bacteroidota* and *Proteobacteria* (Figure 4B). At the genus level, TXOS treatment increased the abundances of *Bifidobacterium* and *Dubosiella* and decreased that of *Faecalibaculum* (Figure 4C). We found that administration of TXOS in mice reduced the abundance of the *Lachnospiraceae* family (Figure 4D), suggesting that the TXOS-mediated regulation of obesity-related metabolism does not seem to be associated with an increased abundance of the *Lachnospiraceae* family. Intervention with TXOS alleviated the endotoxaemia caused by intestinal flora imbalance and restored the composition of intestinal flora to a certain extent, especially by increasing the abundance of *B. pseudolongum*. Moreover, *F. rodentium* and *L. johnsonii* proliferated strongly in the TXOS-treated groups (Figure 4E). As functional oligosaccharides, TXOS protected the intestinal barrier and promoted the growth of beneficial bacteria such as *B. pseudolongum*, *A. muciniphila*, *D. newyorkensis*, and *Bacteroides acidifaciens*, alleviating the intestinal flora imbalance (Figure 5B). TXOS treatment significantly increased the levels of *B. acidifaciens* and *A. muciniphila* compared to those in the ND, HFD, and HFDA groups (Figure 5B), indicating an association between the gut microbiota and liver injury phenotype. The relative abundances of *Akkermansia muciniphila*, *Dubosiella newyorkensis*, and *Bacteroides acidifaclens* increased significantly (* *p*-value < 0.05) in the TXOS treatment group mice. Moreover, TXOS treatment reduced the abundances of opportunistic pathogen species, such as *K. variicola* and *Romboutsia ilealis* (Figure 5C). These results indicated that TXOS treatment modulated the gut microbiota of obese mice, resulting in a microbiota composition that was similar to that of the ND group mice. The data strongly suggest that TXOS does not simply restore the normal microflora, but instead it induces significant gut alterations that are linked to its beneficial effect.

We evaluated the functions of the gut microbiota using the FAPROTAX (Functional Annotation of Prokaryotic Taxa) database, and 2259 OTUs were classified into 30 functional groups (Figure 6A). Differences in bacterial functions between the six groups were investigated. “human_gut”, “mammal_gut”, and “fermentation” were different functional groups classified based on the OTUs in the FAPROTAX database, representing certain types of functions. The “human_gut” includes *Blautia*, *Bacteroides*, *Akkermansia*, *and Bifidobacterium*, and “mammal_gut” mainly refers to several kinds of *Catellicoccus*, *Ruminococcaceae*, and *Helicobacter*. The abundances of bacterial groups associated with aromatic compound degradation, xylanolysis, and fumarate respiration (by members affiliated with *Actinobacteria*, *Firmicutes*, *Gammaproteobacteria*, *Alphaproteobacteria*, *Bacteroidia*, and *Deltaproteobacteria*) in the ND and HFDH groups were significantly higher than those in the HFD, HFDA, HFDM, and HFDL groups (* *p*-value < 0.05). Bacterial groups associated with fermentation (primarily by members affiliated with *Deltaproteobacteria*, *Enterobacteriales*, and *Actinobacteria*) were enriched in the TXOS treatment groups (* *p*-value < 0.05). The HFD group exhibited the highest enrichment of 10 functions, including photoheterotrophy and nitrite respiration (primarily by members affiliated with Gammaproteobacteria and Firmicutes). The results of the FAPROTAX function prediction are consistent with the LEfSe analysis, and the TXOS treatment group was mainly used to maintain the functions of the mammal_gut and the human_gut.

### 3.5. Correlation between the Obesity Parameters and Gut Microbiota

Spearman’s correlation analysis between the obesity parameters and hepatic fatty acid levels in different groups was further performed (Figure 6B and Appendix A). The C18:2(n-6) level was negatively correlated with the levels of TC and HDL-C (* *p*-value < 0.05, r = −0.60, −0.62). The C18:1(n-12) T level was positively correlated with the levels of TC, SREBP1c, PPARG, and FASN (* *p*-value < 0.05, r = 0.63, 0.63, 0.77, 0.69). The levels of C14:0 and C16:0 were negatively correlated with the HDL-C level (* *p*-value < 0.05, r = −0.67, −0.62). The level of LXR was positively correlated with the levels of C14:1, C24:0, and C18:3(n-3) (* *p*-value < 0.05, r = 0.63, 0.75, 0.74). The SOD level was negatively correlated with the C19:1(n-9) T level (* *p*-value < 0.05, r = −0.66).

The HDL-C level in mouse serum was positively correlated with the abundances of *B. pseudolongum* and *D. newyorkensis* (* *p*-value < 0.05, r = 0.74, 0.59), but it was negatively correlated with the abundances of *Lactobacillus murinus* and *Lachnospiraceae bacterium* 28-4 (* *p*-value < 0.05, r = −0.67, −0.72). Moreover, the abundance of *B. pseudolongum* was positively correlated with those of *A. muciniphila* and *D. newyorkensis* (* *p*-value < 0.05, r = 0.64, 0.59). Interestingly, the reduced abundances of some opportunistic pathogen species were negatively correlated with the *B. pseudolongum* abundance (Figure 6B and Appendix A). Other significant correlations between microbial species and genes are presented in Appendix A. Notably, the abundances of *B. pseudolongum* and *A. muciniphila* were positively correlated with the level of PPAR a, which was enriched by TXOS treatment. In addition, probiotics were commonly found in the gut of obese mice. In particular, the abundances of *Lactobacillus johnsonii*, *Faecalibaculum rodentium*, and *Lactococcus lactis* were associated with the levels of MDA and PPARG.

The levels of several typical gut microflora-related fatty acids were highly correlated with the abundances of specific gut bacteria, and a functional correlation between the gut microbiota and liver metabolites was demonstrated (Appendix A). The level of caprylic acid (C8:0), which decreased 2.45-fold in the TXOS-treated mice, was positively correlated with the abundances of *K. variicola* and *R. ilealis* but negatively correlated with the abundances of *B. pseudolongum* and *A. muciniphila*. Likewise, the level of a-linolenic acid (C18:3(n-3)) was positively correlated with the abundances of *K. variicola* and *R. ilealis*. Thus, TXOS treatment induced a significant taxonomic perturbation in the gut microbiota, which, in turn, substantially altered the metabolomic profile, as evidenced by changes in diverse gut microflora-related metabolites. These results suggest that TXOS treatment attenuates metabolic disorders via the gut–liver axis in mice with diet-induced obesity.

## 4. Discussion

Functional oligosaccharides exhibit obesity-reducing effects [15,26]. The major mechanisms have been found to promote the growth of inherited gut microbes or probiotics [20]. In the present study, we demonstrated a range of beneficial effects of TXOS supplementation in obese mice through the gut microbiota–liver axis. To the best of our knowledge, this study is the first to report that TXOS supplementation alleviates metabolic disorders in C57/BL 6J mice with HFD-induced obesity. Several studies have shown that unsaturated alginate oligosaccharides (UAOSs) have an anti-obesity effect and a dose-dependent effect on weight loss [27]. As shown in Table 1, the effect of TXOS (4.8 g/kg per day) on reducing bodyweight gain of mice with HFD-induced obesity was similar to that of orlistat. In addition, there were no significant differences between the HFDM, HFDL, and HFD treatment groups (Table 1), indicating that the TXOS-induced alteration in bodyweight was dose-dependent. The results are similar to the ulvan oligosaccharide (UO) treatment of C57BL/6 J mice, with no significant changes in bodyweight gain detected in the UO groups [28]. The C57BL6J study used a diet-induced obesity model (DIO) that resembles the human condition arguably more closely. However, rodents are more resistant to obesity than humans overall, so in order for obesity to be achieved, a higher caloric imbalance needs to be inflicted and for longer. This situation may induce further oxidative stress and inflammation, leading to an increased tendency for several diseases, metabolic and otherwise [29,30].

Metabolomics has been widely used to screen and identify biomarkers associated with various diseases [31]. Saturated fatty acids are specified by the number of carbon atoms, and medium-chain fatty acids (MCFAs) have been shown to have protective effects on glucose homeostasis during high-fat overfeeding in rodents, and are mainly taken up and metabolized in the liver [32]. MCFAs are directly absorbed into the liver via the portal vein and are used as an energy source without the carnitine transport system for mitochondrial entry [33]. They suppress fat deposition through enhanced thermogenesis and fat oxidation in animal and human subjects. Moreover, hexanoic (C6:0), octanoic (C8:0), capric (C10:0), and lauric (C12:0) acids have received particular attention in metabolic studies [34]. Decanoic acid (caprylic acid) has anti-inflammatory activity and can also alleviate cyclophosphamide-induced oxidative stress by increasing SOD and GSH-px levels. As a crucial organ in overall lipid homeostasis, the liver plays a critical role in many metabolic diseases. In this study, targeted metabolomic analysis was performed to screen fatty acids, and 47 fatty acids were identified (Appendix A). The level of C10:0 increased from 0.83 ± 0.20 μg/g to 0.92 ± 0.24 μg/g, which might be related to the protection of mice from HFD-induced liver injury. Long-chain fatty acids (LCFAs) serve as permanently attached mitochondrial uncoupling protein 1 (UCP1) substrates that help to carry H^+^ via UCP1 [35]. A negative energy balance created by MCFAs may promote fat oxidation and weight loss in individuals with obesity [33]. The CoA derivatives of C8:0 and Cl2:0 exhibited the strongest activity, followed by C16:0 and C24:0.

These fatty acids were involved in five metabolic pathways (Figure 2B). Furthermore, metabolic pathways, including α-linolenic acid, linolenic acid, and lipoic metabolism, were disturbed in obese mice and were markedly improved by TXOS. The MCFAs C8:0 and C12:0 were identified as biomarkers associated with obesity in mice, and decreased weight gain during TXOS treatment was observed. The results indicated that the TXOS-altered pathways were indeed consistent with an increase in the levels of fatty acid oxidation pathways, including the oxidation of fatty acids and lipoic and linolenic acid metabolism. However, fatty acid gavage was not measured and, therefore, the capacity of MCFAs to act as an agent for the treatment of obesity remains to be determined. The precursors of poly-unsaturated fatty acids (PUFAs) are linoleic acid (18:2 n-6) and α-linolenic acid (18:3 n-3), which are also precursors of both pro- and anti-inflammatory mediators [36]. The levels of C22:5(n-6) and C20:3(n-3) were higher in the HFDH group mice than in the HFD group mice (Appendix A). The gut microbiota affected the metabolism of hepatic fatty acids (Figure 2C,D), which is clearly supported by a previous study [20]. Thus, alterations in gut microbial diversity significantly contributed to fatty acid levels in this study.

Many functional oligosaccharides have been shown to have anti-obesity effects [27,37]. However, oligofructose did not prevent obesity in mice consuming an HFD [38]. The Chao1 index was used to estimate the richness and the Shannon index was used to estimate the diversity of gut microbiota. The treatment with prebiotics would increase or show effects on the richness and diversity of gut microbiota. The infant formula with galacto-oligosaccharides increased the abundance of bifidobacteria and resulted in a reduced α-diversity of microbiota [39]. A previous report demonstrated that the α-diversity values for the bacterial community analysis with the Chao 1 and Shannon indexes in Ginseng extract-treated mice were significantly lower than those in the control mice [40]. As shown in Figure 3C,D, TXOS treatment would reduce the Chao1 and Shannon indexes of gut microbiota. The results might be due to the limitations of the chosen technology; the 16S rRNA gene amplicon sequencing could not cover 100% of the gut microbiota. Beyond this, TXOS treatment resulted in the proliferation the *Bifidobacterium pseudolongum* and reduced the abundance of opportunistic pathogen species. The gut microbiota is crucial for the development of adipose tissue, and affects overall weight and the evolution of hepatic steatosis. The gut of mice and humans is heavily colonized by different bacteria; however, only two major phyla are dominant: *Firmicutes* and *Bacteroidetes*. Obesity is associated with a reduction in the abundance of bacteria of the phylum *Bacteroidetes* and an increase in the abundance of bacteria of the phylum *Firmicutes* [41].

The dysregulation of fatty acid metabolism in the gut–liver axis, designating the bidirectional relationship between the gut, microbiome, and liver, is closely associated with a range of human diseases, such as metabolic disorders, inflammatory disease, and carcinoma in the gastrointestinal tract and liver [42]. The gut microbiota affects the profiles of particular fatty acids and glycerophospholipids in the liver and plasma [20]. In this study, a positive relationship between secondary bile acids such as caprylic acid and pathogenic bacteria such as *K. variicola* and *R. ilealis* was observed (Appendix A). Thus, TXOS might attenuate metabolic disorders by modulating fatty acid pathways and reducing toxic metabolites of the gut flora. In addition, Spearman’s correlation analysis of clinical factors of the intestinal flora showed that the *Bifidobacterium* abundance was significantly positively correlated with the HDL-C level, strongly indicating that the TXOS dose can be effectively correlated with the HDL-C level. The intestinal flora is regulated to increase the abundance of beneficial bacteria and reduce the abundance of harmful bacteria, thereby controlling obesity.

## 5. Conclusions

TXOS could reduce obesity indicators in obese mice, control bodyweight, and improve dyslipidaemia. They also alleviated liver lipid metabolism abnormalities. The underlying mechanism may involve the regulation of intestinal bacteria and the changing levels of hepatic fatty acid. These alterations in host lipid metabolism might influence not only obesity-related parameters but also anti-inflammatory molecules. We will further analyse and verify which pathways are involved in these phenotypic changes via Western blotting and knockout mice in future studies.

## Figures and Tables

**Figure 1 foods-12-01382-f001:**
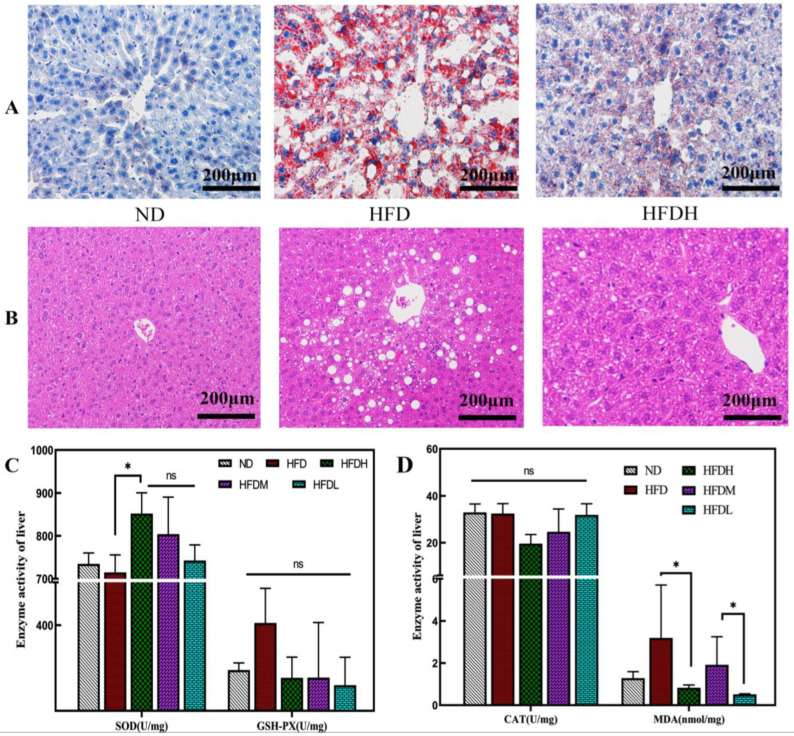
Regulatory effect of TXOS on liver fat accumulation and antioxidant parameters in HFD−fed mice. The data are shown as the means ± SDs. Means with different letters are significantly different (* *p*-value < 0.05). (**A**) Liver stained with oil red O (200×); (**B**) haematoxylin-eosin staining of liver (200×); (**C**) activities of SOD and GSH−PX in the liver of HFD−fed mice after supplementation with TXOS; (**D**) activities of CAT and MDA in the liver of HFD−fed mice after supplementation with TXOS.

**Figure 2 foods-12-01382-f002:**
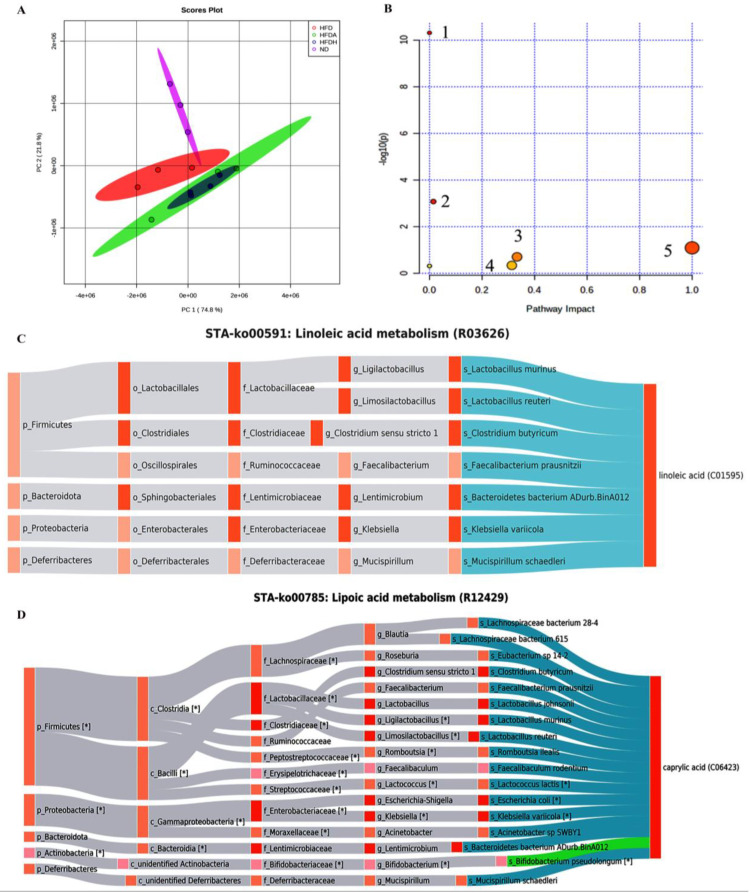
Hepatic fatty acid pathways and Sankey network of potential biomarkers. (**A**) The OPLS−DA plot revealed a significant difference in the abundance of liver fatty acids between different groups. (**B**) Fatty acid pathways of potential biomarkers. 1. Biosynthesis of unsaturated fatty acids; 2. fatty acid biosynthesis; 3. α−linolenic acid metabolism; 4. lipoic acid metabolism; 5. linolenic acid metabolism. (**C**) Sankey network of linolenic acid metabolism and the gut microbiota. (**D**) Sankey network of lipoic acid metabolism and the gut microbiota. Asterisks (*) indicate statistically significant correlations with metabolites. The pink/orange colour of nodes indicates up/down regulation. The pink/orange bands indicate the positive/negative correlations with metabolites. The dark pink/orange colour indicates statistical significance at * *p* < 0.05.

**Figure 3 foods-12-01382-f003:**
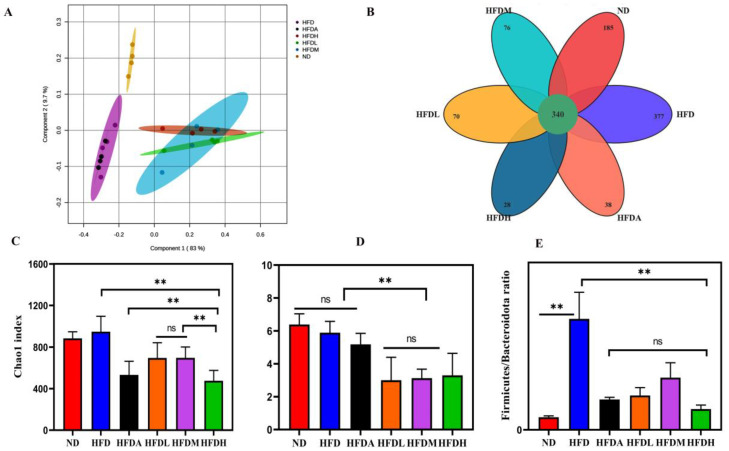
TXOS alleviated HFD−induced gut dysbiosis. The microbiota composition of chow−fed mice and HFD−fed mice treated with TXOS (1.6, 3.2, and 4.8 g/kg per day) was analysed using 16S rRNA pyrosequencing (*n* = 4 for each group). Normal diet group (ND); high−fat diet group (HFD); high−fat diet + orlistat group (HFDA); high−fat diet + high dose of TXOS group (HFDH); high−fat diet + moderate dose of TXOS group (HFDM); high−fat diet + low dose of TXOS group (HFDL). (**A**) The PLS−DA plot revealed a significant difference in the abundance of OTUs between different groups. (**B**) The OTU data of the different groups. The meaning of “340” in the middle of the Venn diagram is the common OTUs found in six different treatment groups. (**C**) Chao1 index of different groups. (**D**) Shannon index of different groups. (**E**) Firmicutes to Bacteroidetes ratio in the indicated groups. (** *p*-value < 0.01). The analyses were conducted using R software version 3.3.1.

**Figure 4 foods-12-01382-f004:**
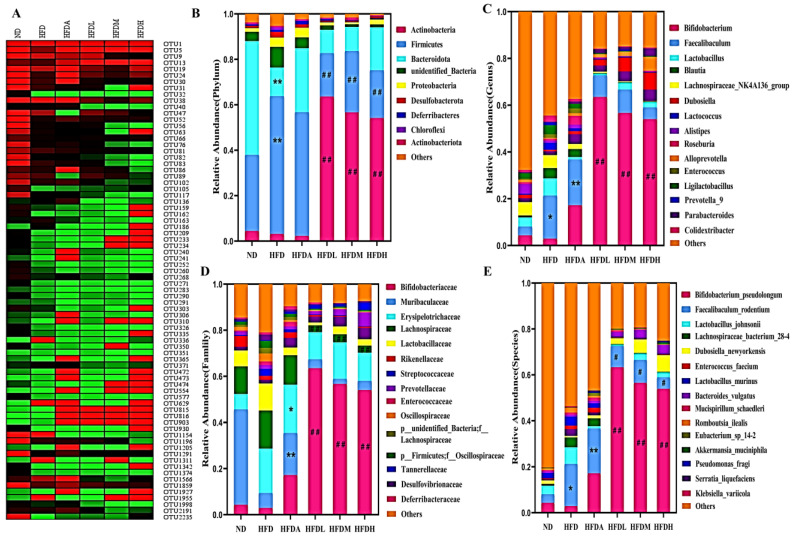
TXOS alleviated HFD-induced gut dysbiosis. Microbiota composition of chow-fed mice and HFD mice treated with TXOS (1.6, 3.2, and 4.8 g/kg per day) were analysed using 16S rRNA pyrosequencing (n = 4 for each group). (**A**) Heatmap of the OTUs in the chow group altered by HFD responding to TXOS treatment. (**B**–**E**) Effect of TXOS on the microbiota profiles and relative abundance at the different levels (phylum, class, order and family) (* *p*-value < 0.05), (** *p*-value < 0.01), comparisons between the ND and HFD groups; (^#^ *p*-value < 0.05, ^##^ *p*-value < 0.01), comparison between the HFD and TXOS groups. The analyses were conducted using R software version 3.3.1.

**Figure 5 foods-12-01382-f005:**
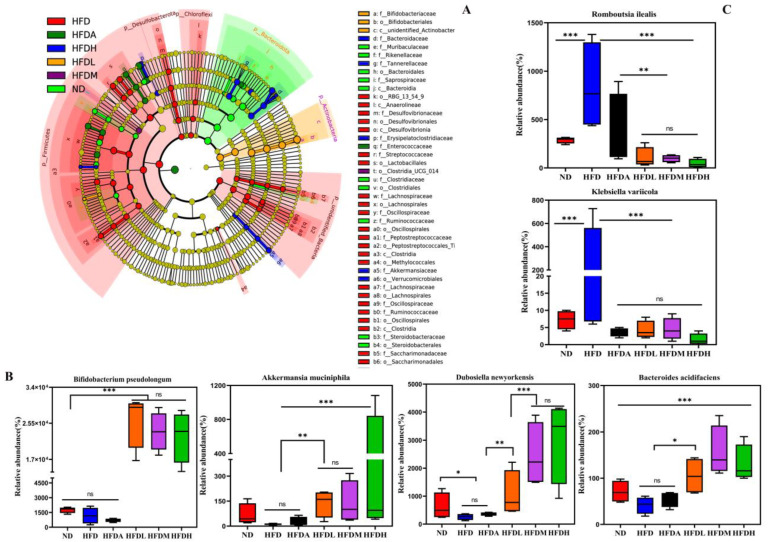
Species with significantly increased and decreased abundances in TXOS-treated mice. (**A**) Taxonomic differences in the faecal microbiota between six groups; cladogram obtained through LEfSe. (**B**) Species with significantly increased abundances in TXOS-treated mice (top four). (**C**) Species with significantly decreased abundances in TXOS-treated mice (top two). The bars indicate the means ± SDs. Statistical analyses: *t*-test. * *p*-value < 0.05, ** *p*-value < 0.01, *** *p*-value < 0.001. ns, not significant. Normal diet group (ND); high-fat diet group (HFD); high-fat diet + orlistat group (HFDA); high-fat diet + high dose of TXOS group (HFDH); high-fat diet + moderate dose of TXOS group (HFDM); high-fat diet + low dose of TXOS group (HFDL). Asterisks (*) indicate statistically significant correlations with different groups.

**Figure 6 foods-12-01382-f006:**
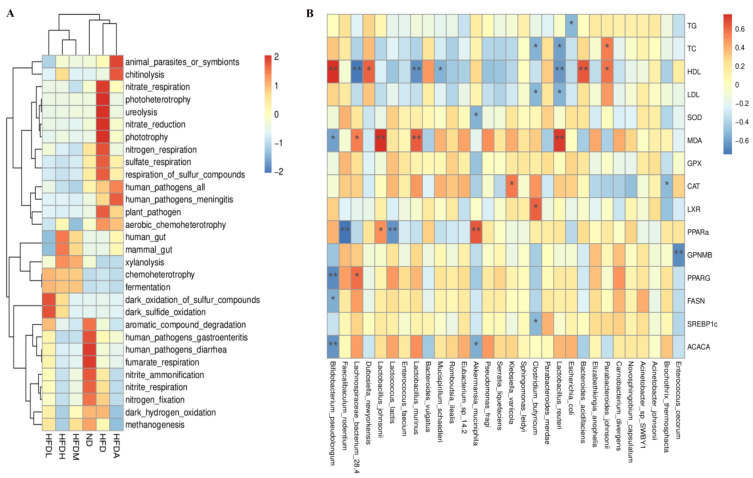
Variations in microbial community compositions and functional characteristics as well as Spearman’s correlation of obesity for the microbiota and parameters. (**A**) FAPROTAX heatmap of the groups (top > 30). (**B**) Spearman’s correlation of obesity for the microbiota, serum, and liver parameters, * *p*-value < 0.05, ** *p*-value < 0.01.

**Table 1 foods-12-01382-t001:** Effect of TXOS on the bodyweight and organ weight of mice with HFD-induced obesity.

Group	Initial Weight/g	Final Weight/g	Weight Gain/g	Perirenal Fat/g	Epididymal Fat/g	Subcutaneous Fat/g	Food Intake/kg
ND	20.12 ± 0.60	26.98 ± 1.16	6.86 ± 0.82	0.21 ± 0.28	0.78 ± 0.68	0.36 ± 0.35	3.56
HFD	20.04 ± 0.68	42.58 ± 7.37 ^##^	22.54 ± 6.96 ^##^	1.04 ± 0.32 ^##^	2.63 ± 0.88 ^##^	1.83 ± 0.51 ^##^	2.03 ^##^
HFDH	19.86 ± 1.00	37.15 ± 7.43 *	17.29 ± 7.82 *	0.83 ± 0.35	2.08 ± 1.02	1.76 ± 1.01	2.03
HFDM	19.78 ± 1.61	40.46 ± 8.08	20.69 ± 7.14	0.78 ± 0.36	2.48 ± 0.66	1.96 ± 1.03	2.39
HFDL	20.05 ± 0.83	43.60 ± 3.90	23.55 ± 3.52	0.87 ± 0.41	2.28 ± 0.76	1.50 ± 0.74	2.30
HFDA	19.41 ± 0.38	33.40 ± 5.65 **	14.00 ± 5.75 **	0.60 ± 0.28 *	1.66 ± 0.72 *	0.84 ± 0.30 *	2.24

Note: normal diet group (ND); high−fat diet group (HFD); high−fat diet + orlistat group (HFDA); high−fat diet + high dose of TXOS group (HFDH); high−fat diet + moderate dose of TXOS group (HFDM); high−fat diet + low dose of TXOS group (HFDL). * Compared with the HFD group, the difference was significant (*p*-value < 0.05). ** Compared with the HFD group, the difference was extremely significant (*p*-value < 0.01). ^##^ Compared with the ND group, the difference was extremely significant (*p*-value < 0.01).

**Table 2 foods-12-01382-t002:** Change trends of the identified potential biomarkers in obese mice treated with TXOS.

Num.	Compound	Formula	Abbreviation	RT (min)	HFD/ND	HFDA	HFDH	Pathway
1	Caprylic acid	C_8_H_16_O_2_	C8:0	3.50	↓ ^###^		↓ ***	Fatty acid biosynthesis
2	Dodecanoic acid	C_12_H_24_O_2_	C12:0	5.11		↓ **	↓ **	Fatty acid biosynthesis
3	Tridecanoic acid	C_13_H_26_O_2_	C13:0	5.65			↓ *	Fatty acids and conjugates
4	Tetradecanoic acid	C_14_H_28_O_2_	C14:0	6.23			↓ **	Fatty acid biosynthesis
5	Myristelaidic acid	C_14_H_26_O_2_	C14:1T	5.47	↑ ^##^	↓ **	↓ **	
6	Palmitelaidic acid	C_16_H_30_O_2_	C16:1T	6.55	↑ ^#^		↓ *	
7	Linoleic acid	C_18_H_32_O_2_	C18:2(n-6)	6.66			↓ *	Linolenic acid metabolism
8	Linoelaidic acid	C_18_H_32_O_2_	C18:2(n-6) T	6.95		↓ ***	↓***	
9	a-Linolenic acid	C_18_H_30_O_2_	C18:3(n-3)	5.93		↓ **	↓ **	α-Linolenic acid metabolism
10	cis-5,8,11,14,17-Eicosapentaenoic acid	C_20_H_30_O_2_	C20:5	5.77	↑ ^###^	↓ ***	↓ ***	Biosynthesis of unsaturated fatty acids
11	cis-7,10,13,16,19-Docosapentaenoic acid	C_22_H_34_O_2_	C22:5(n-3)	6.59			↓ *	Biosynthesis of unsaturated fatty acids
12	Tetracosanoic acid	C_24_H_48_O_2_	C24:0	10.86		↓ **	↓ **	Biosynthesis of unsaturated fatty acids
13	Myristoleic acid	C_14_H_26_O_2_	C14:1	5.32		↓ ***		Fatty acids and conjugates
14	Heneicosanoic acid	C_21_H_42_O_2_	C21:0	9.76	↓ ^##^			
15	Nervonic acid	C_24_H_46_O_2_	C24:1	10.08	↓ ^#^			Biosynthesis of unsaturated fatty acids
16	Petroselinic acid	C_18_H_34_O_2_	C18:1(n-12)	7.66	↑ ^##^			Fatty acids and conjugates

The levels of potential biomarkers are labelled with an increase (↑) or a decrease (↓). RT, retention time. “^#^” indicates a significant change in HFD vs. ND (^###^ *p*-value < 0.001; ^##^
*p*-value < 0.01; ^#^
*p*-value < 0.05); “*” indicates a significant change in different treatment groups vs. the HFD group (*** *p*-value < 0.001; ** *p*-value < 0.01; * *p*-value < 0.05). ND, normal diet group; HFD, high-fat diet group; HFDA, 60 kcal% fat diet + orlistat group; HFDH, 60 kcal% fat diet + high dose of TXOS group (4.8 g/kg).

## Data Availability

The data used to support the findings of this study can be made available by the corresponding author upon request.

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
