# Peer review of "Tamarind Xyloglucan Oligosaccharides Attenuate Metabolic Disorders via the Gut–Liver Axis in Mice with High-Fat-Diet-Induced Obesity"

_foods, 2023, doi:10.3390/foods12071382_

Round 1
Reviewer 1 Report
This is my review on manuscript entitled: Tamarind xyloglucan oligosaccharides attenuate metabolic disorders via the gut-liver axis in mice with high-fat diet-induced obesity.
Introduction presents all the necessary information on the matter. Tamarind seed polysaccharides have indeed potential prebiotic properties based on literature and therefore it is interesting to study their restoration properties regarding the gut dysbiosis.
Materials and methods are thoroughly described. All the necessary approvals regarding the animals were provided. All the laboratory analyses are satisfyingly presented. Statistical analysis is the right too.
Figures 1 and 2 are rather interesting. I would suggest the authors to include bar scales in μm (micrometers) on the red O and HE images (figure 1).
Figures 3 and 4 are important bat rather large and obscure at some points. An improvement of quality should be performed.
Discussion is supported by the results and is well written. Authors should add conclusions (separate the last paragraph from discussion) and implications for further study.
Reviewer 2 Report
Tamarind xyloglucan oligosaccharides attenuate metabolic disorders via the gut-liver axis in mice with high-fat diet-induced obesity
General impression
This is an interesting work looking into the bioactivities of tamarind xyloglucan (TXOS), a putative oligosaccharide-based prebiotic in obesity induced by high-fat diet (HFD). To a certain extent, the supplementation of TXOS prevented the onset of a myriad of HFD-induced metabolic abnormalities including, weight gain, dyslipidemia, ectopic fat deposition in the livers, and more importantly the dysregulation of hepatic lipidomes and gut microbiome dysbiosis. The data clearly support the beneficial impacts of tamarind xyloglucan on liver health and metabolism. However, one major shortcoming of the manuscript is its unsatisfactory data presentation, as exemplified by the poor resolution figures and lack of explanation of the figures, rendering it hard to follow through the result section. Improvement is needed on this aspect.
Major comments
1. In the title, results (Lines 367 to 369) and discussion (Lines 374-375), TXOS is said to exert the systemic beneficial effects via the gut-liver axis. However, the team neither shows that changes in gut microbiome precedes the alteration in liver lipidomes, nor establishes that the gut microbes enriched by TXOS could independently ameliorate lipidomic disruption without TXOS. Hence, there is little evidence supporting the proposed mechanism of TXOS. It is advisable to tone down the statements or perform additional experiment to establish the causal relationship between gut microbiome and hepatic lipidome mediated by TXOS.
2. Most of the figures including those in the supplementary figures, have poor resolution. The words are hardly readable without zooming in, especially for Fig 2C-D, 4A, S3. Key details and explanation are missing in the figure legends, making it challenging for data interpretation. Here are some of the issues:
a. Scale bars of Fig 1A-B & S1E are missing.
b. Fig 2A, the colors for HFDA, HFDH and ND are too similar. It is hard to discriminate the groups from each other. It is not sure why HFDL and HFDM were not included.
c. Fig 2B, what does “pathway impact” means? The dot size and color are different for each pathway but their meaning was not clarified in the legend.
d. Fig 2C-D, where do the microbial taxa and abundance come from? The meaning of orange color intensity for each taxa and colors of the links (which include grey, blue and green), were not explained.
e. Fig 3A can be plotted as dot plot to visualize the grouping pattern.
f. Fig 3B, is it supposed to be a Venn diagram? The number (340) in the middle is not explained. This sub-figure is not cited in the manuscript.
g. Fig 4A is too messy. There are too many color coding that overlaps with each other. The words are not clear.
h. Fig 5B , the meaning of the asterisk is not explained. Clustering analysis should be performed to visualize the grouping of gut microbiome and phenotypes.
3. In Section 3.3 Lines 235-236 and Fig 2C-D, hepatic lipidomic data is presented together with gut microbiome changes, but no data on the microbial profile is cited or introduced until later sections. Inclusion of the statements about microbiome is confusing especially when the HFD-induced gut dysbiosis is not established until Section 3.4.
4. In Section 3.4 Lines 291-296, how is it possible to identify the bacterial taxa at species level when 16S rDNA sequencing was used? The bioinformatic pipeline for the gut microbiome analysis is not described.
5. In the abstract and results (Section 3.4; subheading and Lines 297-298) TXOS was said to alleviate HFD-induced gut dysbiosis and restore the normal microbiota composition. However, the claim is not supported by the PCoA analysis (Figure 3A) and functional prediction (Figure 5A) which clearly showed that the gut microbiome of HFDL, -M and -H occupied a unique space in the dissimilarity distance and exerted biological functions different from ND, except for HFDL which is likely due to low TXOS dosage. In Fig S4, it is also very clear that TXOS enriched phylum Actinobacteriota. The data strongly suggests that TXOS does not simply restore the normal microflora, but instead it induces significant gut alterations that are linked to its beneficial effect. It is advisable to revisit the gut microbiome analysis and present the data accurately. Fig S4 should be included as a main figure.
6. Three different dosages of TXOS were used but there is no information about the dose response on any of the measured parameters.
Minor comments
7. The font size keeps changing throughout the manuscript, for example under Method Section 2.2.
8. It should be PPARα, not PPARa. It is an alpha symbol, not alphabet “a”. Please change the typo in the main text, figures and supplementary files.
9. In some figure legends, there are random integers like “344444” and “43344434”. Please omit them.
10. Lines 238-242 should be moved to the following paragraph or omitted.
11. The experiment on the hepatic fatty acids is a lipidomic assessment. Metabolomic study is too general. Please change the wording to lipid/lipidome/lipidomic profile, not metabolite/metabolome/metabolite profile.
12. Full stop should be followed by a space, not the otherwise. Such typo is common in the discussion.
13. In the discussion, Lines 440-465 should be moved to the Result section.
14. In the conclusion (Lines 483-484), “promotion of hepatic fatty acid levels” is inaccurate as not all fatty acids are elevated with TXOS supplementation; some of them were reduced. Please revise the wording to reflect the findings accurately.
Reviewer 3 Report
The manuscript submitted to Foods for publication by Zhu et al., titled: "Tamarind xyloglucan oligosaccharides attenuate metabolic disorders via the gut-liver axis in mice with high-fat diet-induced obesity" is an interesting in vivo study whereby the researchers investigate the effect of a functional oligossachharide on metabolic disorder outcomes through the gut-liver axis. The paper is well structure and nicely organized and fairly easy for the reader to follow as it flows well. The reviewer would like to offer the following points for consideration by the authors for the benefit of the manuscript and its improvement.
1. How was the number of animals determined? (Was there a power calculation, or prior pilot study revealing an optimal minimum number appropriate to capture changes with significant statistical power and confidence?).
2. While the reviewer understands that this is a common approach in terms of experimental design, there seems to be a jump from the mRNA (message/transcription) to the metabolite in the outcome variables assessed, thus the protein/enzyme levels assessment (for some key players eg: regulatory node enzymes for pathways) via Western Blotting for instance which would provide a pipeline and more complete picture as per the intensity of pathway activation at the practical and physiological level is missing. This is not an undo but it would warrant acknowledgment and some discussion.
3. The gut-liver axis is a very interesting approach and in the reviewer's opinion the role of the gut on metabolic disorders should be discussed under the light and through the lens of its complexity. A potentially useful paper to that end is the following: Sikalidis, A.K.; Maykish, A. The Gut Microbiome and Type 2 Diabetes Mellitus: Discussing A Complex Relationship. Biomedicines 2020, 8, 8. https://doi.org/10.3390/biomedicines8010008.
4. The discussion on the model of obesity is also important. The authors have used C57BL6J thus inducing a Diet Induced Obesity model (DIO) which resembles the human condition arguably more closely. However the rodents are more resistant to obesity than humans overall so in order for obesity to be achieved a higher caloric imbalance for longer needs to be inflicted. This situation may induce further oxidative stress and inflammation leading to an increased tendency for several diseases metabolic and otherwise. The distinction would be interesting to be discussed two manuscripts that in tandem discussed the issue using two different obesity models are the following and they could potentially be used in this sense:
- Sikalidis AK, Fitch MD, Fleming SE. (2013) Risk of Colonic Cancer is Not Higher in the Obese Lepob Mouse Model Compared to Lean Littermates. Pathol Oncol Res. 19(4):867-874. doi: 10.1007/s12253-013-9656-7.
- Sikalidis AK, Fitch MD, Fleming SE. (2013) Diet Induced Obesity Increases the Risk of Colonic Tumorigenesis in Mice. Pathol Oncol Res. 19(4):657-666. doi: 10.1007/s12253-013-9626-0.
Good job overall.
Round 2
Reviewer 2 Report
Generally, the authors have addressed most of my major concerns. However, a few minor comments were left out/not adequately addressed.
Point 2e. For Fig 3A, I would expect PCoA1 and PCoA2 to be plotted as x and y axes, respectively in a dot plot. This would allow the visualization of the grouping pattern.
Point 2f. Fig 3B is still not cited in the main text. The meaning of "340" in the middle of the diagram is also not defined. Given that it is a Venn diagram, the overlapping part in the middle should be the common OTU found in all treatment groups. 340 common OTUs are unlikely given that most groups have fewer OTUs than 340.
Point 4. It is still unclear how species level information, such as B. pseudolongum, A. municiphila, D newyorkensis etc. can be detected using 16S-based metagenomics. While some of the sequencing information has been added, the bioinformatic pipeline (e.g. mothur, DADA2 etc.) of 16S metagenomics remains unreported.
Other than that, I have no further comment.
Reviewer 3 Report
The authors have made a reasonable effort in addressing the reviewer's comments.
Author Response
We are grateful for the reviewer 3 comments.